# Activity-mediated accumulation of potassium induces a switch in firing pattern and neuronal excitability type

Susana Andrea Contreras[1,2], Jan-Hendrik Schleimer[1,2], Allan T. Gulledge[3], Susanne Schreiber[1,2¤]*

**1** Institute for Theoretical Biology, Humboldt-University of Berlin, Berlin, Germany, **2** Bernstein Center for Computational Neuroscience Berlin, Berlin, Germany, **3** Molecular and Systems Biology, Geisel School of Medicine at Dartmouth College, Hanover, New Hampshire, United States of America

¤ Current address: Humboldt-University of Berlin, Institute for Theoretical Biology, Berlin, Germany
* s.schreiber@hu-berlin.de

**Data Availability Statement:** The neuron model is described in the manuscript. The code to reproduce the simulations reported in the manuscript can be found in

## Abstract

During normal neuronal activity, ionic concentration gradients across a neuron's membrane are often assumed to be stable. Prolonged spiking activity, however, can reduce transmembrane gradients and affect voltage dynamics. Based on mathematical modeling, we investigated the impact of neuronal activity on ionic concentrations and, consequently, the dynamics of action potential generation. We find that intense spiking activity on the order of a second suffices to induce changes in ionic reversal potentials and to consistently induce a switch from a regular to an intermittent firing mode. This transition is caused by a qualitative alteration in the system's voltage dynamics, mathematically corresponding to a co-dimension-two bifurcation from a saddle-node on invariant cycle (SNIC) to a homoclinic orbit bifurcation (HOM). Our electrophysiological recordings in mouse cortical pyramidal neurons confirm the changes in action potential dynamics predicted by the models: (i) activity-dependent increases in intracellular sodium concentration directly reduce action potential amplitudes, an effect typically attributed solely to sodium channel inactivation; (ii) extracellular potassium accumulation switches action potential generation from tonic firing to intermittently interrupted output. Thus, individual neurons may respond very differently to the same input stimuli, depending on their recent patterns of activity and/or the current brain-state.

## Author summary

Ionic concentrations in the brain are not constant. We show that during intense neuronal activity, they can change on the order of seconds and even switch neuronal spiking patterns under identical stimulation from a regular firing mode to an intermittently interrupted one. Triggered by an accumulation of extracellular potassium, such a transition is caused by a specific, qualitative change in of the neuronal voltage dynamics—a so-called bifurcation—which affects crucial features of action-potential generation and bears consequences for how information is encoded and how neurons behave together in the

https://itbgit.biologie.hu-berlin.de/contreras/activity-mediated_accumulation_potassium_induces_switch_firing. The neuronal recordings can be found in publicly available repository: https://gin.g-node.org/Contreras/Activity-mediated_accumulation_potassium_induces_switch_firing_pattern.

**Funding:** This project has received funding from the European Research Council (ERC) under the European Union's Horizon 2020 research and innovation program (grant agreement No 864243) and the German Ministry of Education and Research (grant number 01GQ1403) to SS, the Deutsche Forschungsgemeinschaft (GRK 1589/2) to SAC, and the National Institutes of Health-National Institute for Mental Health (R01 MH099054) to ATG. The funders had no role in study design, data collection and analysis, decision to publish, or preparation of the manuscript.

**Competing interests:** The authors have declared that no competing interests exist.

network. Also, changes in intracellular sodium can induce measurable effects, like a reduction of spike amplitude that occurs independently of the fast amplitude effects attributed to sodium channel inactivation. Taken together, our results demonstrate that a neuron can respond very differently to the same stimulus, depending on its previous activity or the current brain state. This finding may be particularly relevant when other regulatory mechanisms of ionic homeostasis are challenged, for example, during pathological states of glial impairment or oxygen deprivation. Finally, categorization of cortical neurons as intrinsically bursting or regular spiking may be biased by the ionic concentrations at the time of the observation, highlighting the non-static nature of neuronal dynamics.

## Introduction

Ever since the introduction of Hodgkin-Huxley's famous neuron model for the squid giant axon, the governing equations have been a useful tool to understand the mechanisms of spike generation. The original model assumed fixed ionic concentrations inside and outside the cell, establishing constant driving forces for ionic flux otherwise modulated only by the channels' gating kinetics [1]. In the brain, however, ionic concentrations are not constant, and the ionic composition of the extracellular space varies with behavioral states [2, 3] and as a function of neuronal activity [4, 5].

The concentrations of sodium $[Na^+]$ and potassium $[K^+]$ ions—the two ionic species essential for sodium action potentials—are known to vary in response to neuronal activity *in vitro* and *in vivo* at relatively slow timescales (on the order of seconds). Intracellular sodium concentration has been found to increase with activity in mammalian pyramidal neurons responding to physiologically relevant stimuli (on the order of 3–10 seconds) [5]. In the cat neocortex, the concentration of extracellular potassium can oscillate in correlation with local field potentials (LFPs) during slow wave ($\sim 1$ Hz) sleep [3] or when presenting oscillating graded stimuli to the cat's retina on the order of seconds [4].

A number of simulation studies have analysed the slow ionic concentration dynamics and their equilibria [6–10], portraying mechanistic explanations of the emergence of slow ionic concentration oscillations (0.5–10 cycles per minute). These studies have particularly focused on understanding how ionic homeostasis [7, 8, 10, 11] or stimulus properties [12] may shape ionic concentration equilibria. However, only few have analysed the transient effect of these concentration changes on neuronal excitability [6, 13]. The majority of the aforementioned studies have adopted a static stimulation, except for [12] who showed that a periodic step current injection can drastically alter ionic concentration equilibria. Thus, how stimulus-induced changes in ionic concentration gradients impact ongoing neuronal activity is currently not well understood.

In this study, we use conductance-based models to predict and experimentally test how changes in transmembrane ionic concentration gradients that arise during periods of increased neuronal activity impact action-potential generation. We find that prolonged stimulation ($\sim 10$ seconds) can generate ionic concentration changes substantial enough to modify action potential generation in neurons. Intracellular sodium accumulation in particular alters action-potential amplitude on slow timescales matching the ionic changes—an effect previously attributed primarily to the inactivation of sodium channels [13–16].

Extracellular potassium accumulation, in turn, can qualitatively switch the spike-generating mechanism of type I neurons (i.e., with a smooth increase in firing rate from threshold), thus changing fundamental properties of firing patterns, encoding, and network behaviour.

Mathematically, the transition corresponds to a so-called co-dimension-two bifurcation, at which the spike generating mechanism changes qualitatively from a regular saddle-node on invariant cycle (SNIC), when extracellular potassium concentrations are low, to a homoclinic orbit bifurcation (HOM), when extracellular potassium concentrations become high. The switch in the firing regime most notably results in a transition from regular spiking to a burst-like, intermittently interrupted firing mode in the HOM regime, caused by a so-called bistability of the dynamical system. In the HOM regime, the options of a fixed, resting-like voltage state and regular firing co-exist for the same input levels, resulting in stimulus- and noise-induced switches between both states.

Prolonged electrical activity can, therefore, have significant effects on spiking patterns and neuronal dynamics. We uncover these properties by first dissecting both potassium ion and sodium ion contributions to spike generation and second testing predictions in *in vitro* electrophysiological recordings.

## Results

### Model response to prolonged stimuli

In order to analyze how neurons respond to prolonged stimulation, we examined the temporal evolution of activity-dependent changes in transmembrane ionic gradients and assessed their impact on ongoing neuronal activity. To this end, we implemented a single-neuron, conductance-based model including dynamic ion concentrations (detailed in the Materials and methods section). Ionic gradients determine the equilibrium (Nernst) potentials that in turn influence the driving forces of spike-generating ionic currents. Accumulation of ions over time consequently modifies the Nernst potentials as well as the spike generating currents and therefore also spike generation. A regulation of concentration gradients ($[Na^+]$ and $[K^+]$) is mediated by the Na-K-ATPase pump: an electrogenic active-transporter whose activity intensifies when $[Na^+]_i$ accumulates. Due to its electrogenic nature (changing the net charges across the membrane), activity of the Na-K-ATPase pump affects the membrane potential.

**Noise-free analysis.** First, we investigated the response of the model to a step input current—a typical protocol in patch-clamp experiments. Stimulation of the model for almost 10 seconds (Fig 1A) led to an accumulation of intracellular sodium, $[Na^+]_i$, as well as an increase in extracellular potassium, $[K^+]_o$ (Fig 1B). The concentration changes resulted from the prolonged spiking activity and were indeed substantial enough to alter features of the generated action potentials during the duration of the stimulation protocol. Three major changes that have often been reported in experiments were observed in the model: (a) the emergence of a slow after-hyperpolarization (AHP), (b) adaptation (i.e., a reduction) of spike frequency, and (c) reduction of spike amplitudes (Fig 1A).

a) The slow AHP became visible when the stimulus was set back to baseline and neuronal spiking stopped (Fig 1A). The slow AHP resulted from the hyperpolarising Na-K-ATPase pump current: Na-K-ATPase pump activity was enhanced by the action-potential-driven rise in intracellular sodium concentration because $[Na^+]_i$ accumulation increases the pump activity. When the stimulation ended, the neuron stopped firing and the membrane potential hyperpolarized with respect to the original resting membrane potential due to the transient change in the Na-K-ATPase pump current. As ongoing Na-K-ATPase activity progressively lowered the intracellular sodium concentration back to baseline levels, the hyperpolarization slowly diminished (Fig 1A).

b) Spike frequency adaptation, evident in Fig 1A, also resulted from the activity-dependent increase in Na-K-ATPase current, which effectively reduced the net excitatory drive of the neuron. The model does not contain adaptation currents besides the Na-K-ATPase pump (*e.g.*,

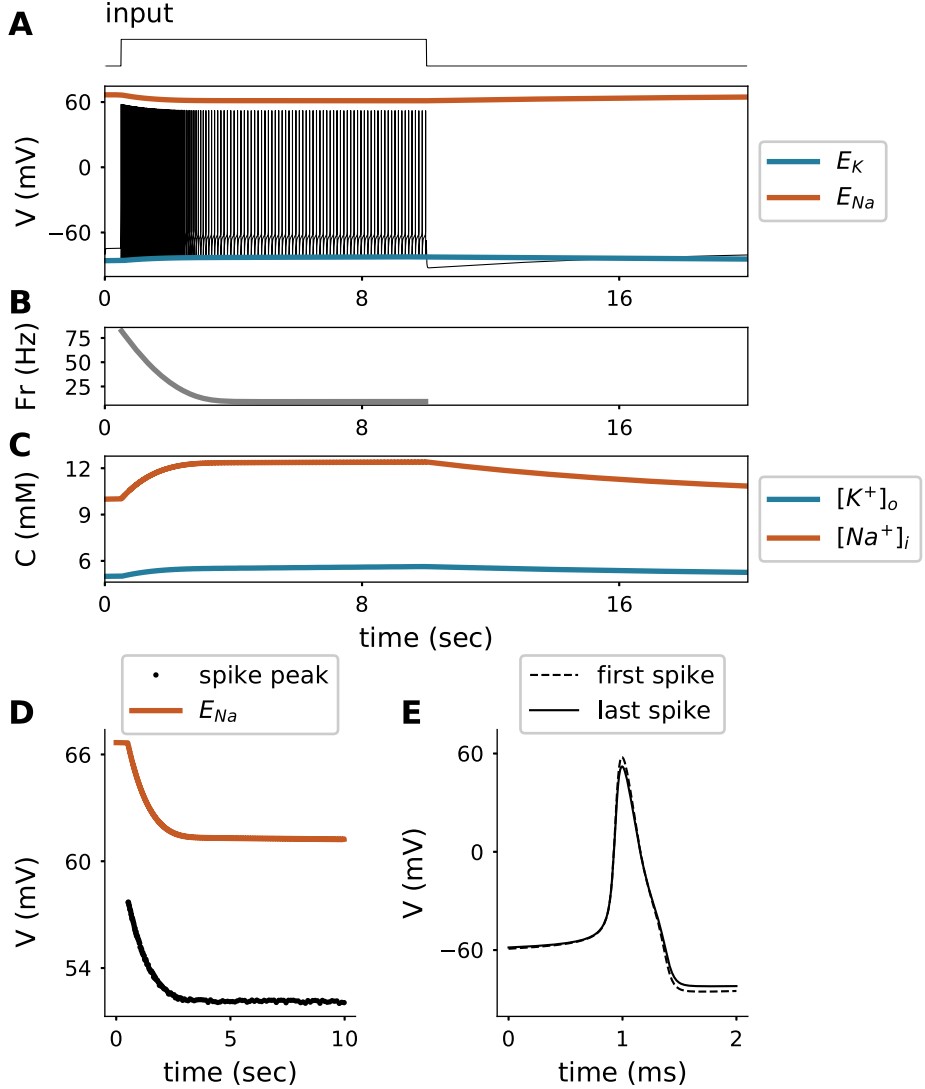

**Fig 1. Response of the model to a step current of 2 $\mu A/cm^2$ input as shown in the top. A**. Voltage trace (V) with reversal potentials for sodium (Orange) and potassium (Blue).**B**. Firing rate (Fr) of neuron model. **C**. Intracellular sodium (Orange) and extracellular potassium (Blue) concentration (C) dynamics. **D**. Spike amplitude and reversal potential for sodium (Orange) for the trace shown in **A**. **E**. First and last spike (peaks aligned).

M-currents). Note that the pump current used in the model is only sensitive to Na⁺, which idealizes the $\alpha3$ isoform of the Na-K-ATPase. The pump $\alpha3$ isoform is negligibly sensitive to K⁺ and the membrane potential (V), but highly sensitive to changes in intracellular sodium concentration over the ranges simulated in this study [17, 18]. The impact of the pump current on spike-frequency adaptation was, however, preserved in models of other pump isoforms.

c) In our model, reductions in spike amplitude were directly related to intracellular sodium accumulation, see Fig 1C. Activity-dependent reduction of action potential amplitude has previously been attributed primarily to Na⁺-channel inactivation during prolonged stimulation [13, 14, 19]. Our simulations, however, demonstrate that the time course of amplitude reduction mirrors the drop in sodium reversal potential (see Fig 1A), which is related to the time course of sodium accumulation (see Fig 1B). The Na⁺-channel inactivation was more than an order of magnitude faster than the timescale of spike-amplitude reduction; the slow spike

amplitude decay was modulated by intracellular sodium. A dynamical system's perspective of this finding and an experimental confirmation are presented in the next sections.

**Analysis in the presence of noise.** The results so far reflect idealized model responses in the absence of noise. To include the stochasticity of synaptic inputs that is typical for many neurons in the central nervous system, we next added coloured noise with wide sense stationary statistics to the input current—a useful exercise that reveals an interesting property in the response that was masked in the noise-free case discussed above.

Stimulating the model again with a step current in the presence of an additional colored noise component (Fig 2), the model neuron's response during the first second was comparable to the noise-free case presented before: $[Na^+]_i$ and $[K^+]_o$ accumulated (resulting in changes in the reversal potentials $E_{Na}$ and $E_K$); spike frequency adaptation, and spike-amplitude reduction were observed (compare to Fig 1A). Surprisingly, after the first second of stimulation, the response exhibited a sudden transition from regular spiking to an intermittently-interrupted, burst-like firing mode. Note that the stimulus statistics are in a wide-sense stationary such that there was no qualitative change in the stimulus during the simulation duration. This means that the qualitative switch in the firing pattern must arise from a bifurcation in the neuron's dynamics. The switch in firing pattern occurred 1.2 seconds after stimulus onset, a time scale that largely exceeds the time scale associated with the dynamics of spike-generating conductances (which are about two orders of magnitude faster). Yet this time scale matches the time scale of changes in ionic concentrations, suggesting that the switch is causally related to the ion accumulation. Ion accumulation influences spike generation by changing ionic reversal potentials and engaging the electrogenic Na-K-ATPase.

**Separating the fast and the slow dynamics.** To disentangle the origin of the transition, in a next step, the fast spike-generating dynamics were separated from the slow ionic concentration dynamics using a slow-fast analysis. To this end, we systematically analyzed the fast system with fixed ionic concentrations (i.e., constant values of the slow concentration variables). The latter, however, were chosen from "snapshots" of the values that the concentrations had exhibited in the full system (where concentrations were varying). This approach allowed us to systematically determine how ionic concentration changes shaped the ongoing properties of the fast sub-system. Time scale separation is valid because ionic concentration changes were much slower ($\sim$ seconds) than the spike generating currents ($\sim$ milliseconds) (see the Materials and methods section).

Analysis of the slow-fast system revealed that the qualitatively different spiking response was triggered by a switch in the dynamics of action-potential generation. Mathematically, the model started out in a setting where spiking is initiated via a saddle-node on invariant circle bifurcation (SNIC) [20]. Models with SNIC dynamics can be distinguished by their fixed points, that form an S-shape curve when projected in the V-$I_{app}$ plane [21]. This type of dynamics is characterized by the existence of a unique stable attractor for each input level, i.e., only one, well-defined state that the system converges to. For low inputs this is a fixed point (i.e., the resting state) while for high inputs it is a limit cyle attractor (i.e., the regular spiking state). In models with fixed intra- and extracellular ionic concentrations, this type of dynamics would persist as long as cellular properties remain constant, i.e., across the whole stimulation period. Alterations in the level of ionic concentrations (and hence their transmembrane gradients in terms of reversal potentials), however, can qualitatively switch the dynamics to a different spike-generating bifurcation. A switch in the spike generating bifurcation can be perceived in some qualitative characteristic features of spike trains. Such a transition can, for example, be reflected in an increased or decreased number of attractors. Indeed, when monitoring the number of stable attractors of the corresponding fast system at each point in time, their number changes exactly at the ionic concentrations reached at 1.2 seconds. Here, an

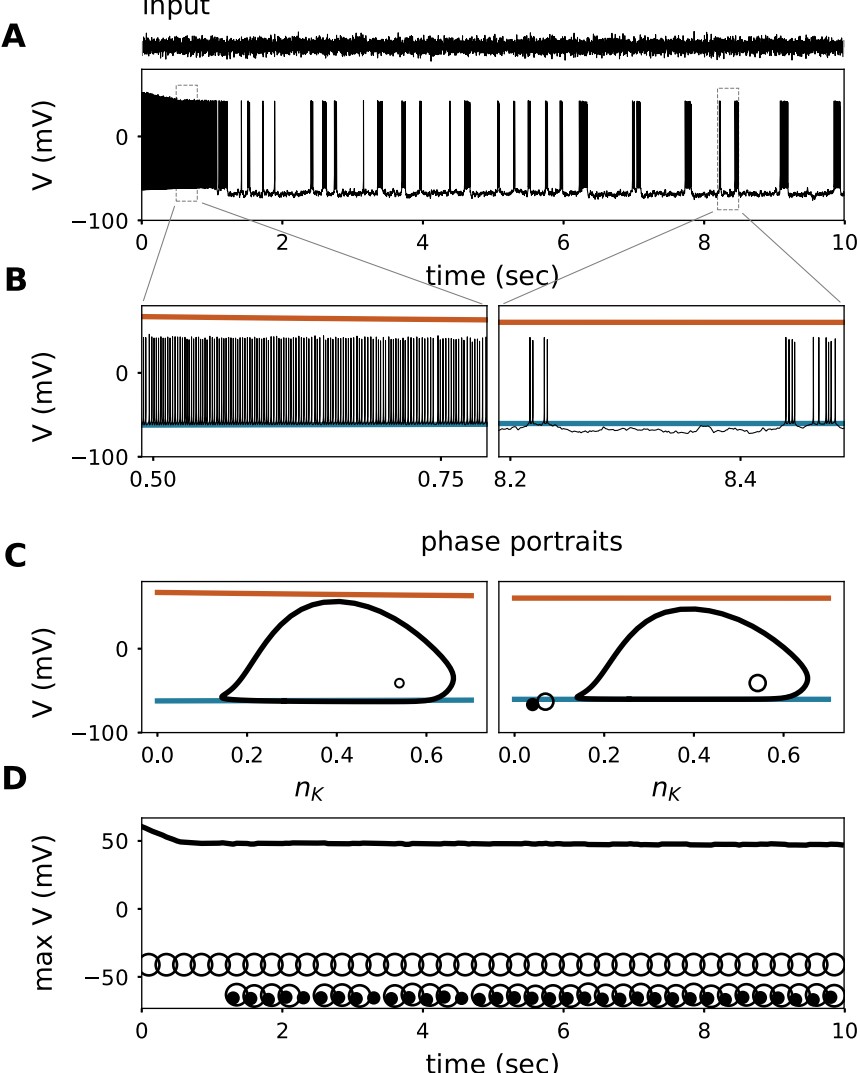

**Fig 2. Response of the model to a step current with colored noise filtered at 500Hz (mean input is $1\mu A/cm^2$ and standard deviation $1.05\mu A/cm^2$, shown above). A**. Voltage trace (membrane potential) of the model responding to a noisy step current (Top of panel). **B**. Zoom of the voltage trace from panel A at the beginning and towards the end of stimulation showing the evolution of the reversal potentials for sodium (orange) and potassium (blue), as in Fig 1. **C**. Phase portraits of the steady state of the fast spike generating sub-system, when imposing the average reversal potentials for sodium (orange) and potassium (blue) of panel B as parameters. Vertical axes show the voltage and the horizontal axes show the potassium current gating-variable ($n_K$). Empty dots are the unstable nodes, filled dots the stable nodes, and the orbits are stable limit cycles. **D**. Evolution of the maximum voltage of the system attractors. Empty dots represent the unstable nodes, filled dots the stable nodes, and the black line denotes the maximum voltage of the stable limit cycles (action potential peak).

additional stable fixed point (i.e., a stable voltage) appears in parallel to the spiking mode for the same size of input current; the system becomes bistable. Which of the two attractors (regular spiking or a fixed voltage) the system converges to depends on the initial conditions and/or noise in the system. In Fig 1, initial conditions are such that the neuron keeps up regular spiking because concentration changes are not substantial enough to reach the switch in spiking dynamics (the emergence of an additional fixed point attractor). When ionic concentration changes are substantial enough they reach the switching point, and the system displays bistable

dynamics (at $\sim 1.2$ sec). In the presence of noise (like in Fig 2) the system continuously receives perturbations and, therefore, when entering the bistable dynamics only temporarily settles onto one of the two attractors before being kicked into the other one. This dynamical state results in a long-lasting, stochastic back-and-forth between periods of spiking and silence (Fig 2). The transition from single attractor to bistability $\sim 1.2$ seconds after stimulation onset is confirmed in the corresponding phase portraits of the fast system (Fig 2**C**).

Bistable *vs.* uni-stable states lead to qualitatively very different responses. The natural question that follows is: what generates the bistability? Mathematically, the bistability is caused by the emergence of a separatrix attached to a saddle point, i.e., a trajectory in phase space that separates the so-called basins of attraction of the two attractors. Depending on which side of the separatrix the system is located at a given point in time, it will converge towards the respective attractor unless noise or an input fluctuation kick the system across the separatrix to the other side. Dynamics in this region are strongly affected by the reversal potential for potassium (see Fig 2**C**). Therefore, we next systematically explored the effects of extracellular potassium on the fast system.

## Consequences of extracellular potassium accumulation

**Dynamical systems analysis.** We analyzed the dynamics of the fast system (i.e., the neuron model with fixed ionic concentrations) for different values of extracellular potassium. Fig 3 shows the resulting two-parameter bifurcation diagram, which depicts the dynamical state as a function of extracellular potassium concentration and size of the applied input current (for details see the Materials and methods section). Four different dynamical regimes can be found: a silent subthreshold state, a regularly spiking state, a bistable state, and a silent state of depolarization block (when the model is depolarized so strongly that spiking cannot occur any more).

Let's look at the diagram (Fig 3) in more detail, starting at lower extracellular potassium values (*i.e.*, the bottom of the diagram). Depending on the input strength, the system here either remains subthreshold or exhibits regular firing. The transition to spiking corresponds to a SNIC bifurcation (See Supplementary material S1 Fig). When elevating the levels of extracellular potassium (to $\sim 12$ mM), the situation changes. Here, an additional (bistable) region appears between the subthreshold and the regular spiking areas. The transition is marked by a codimension-two bifurcation called a saddle-node loop (SNL) [22]. The width of the bistable region increases for higher values of extracellular potassium concentration (dashed lines in Fig 3). The firing threshold corresponds to the left border of the bistable region. The transition to spiking now corresponds to a HOM bifurcation. In the bistable zone, in the presence of noise, intermittently-interrupted firing can be observed. Moreover, at elevated extracellular potassium values, the depolarization block (as the name suggests, usually occurring at very large depolarization levels) can be observed at progressively lower input currents. At very high extracellular potassium values, it directly borders the bistable zone.

The full system (with variable concentrations and pump activity), as analyzed in Fig 2, "lives" in the bottom part of the bifurcation diagram (Fig 3) at the onset of the stimulation, as here the values of extracellular potassium are moderate. Over time, extracellular potassium accumulates and the switching point to HOM dynamics is passed. Here, the bistable range is entered and the burst-like, intermittently interrupted firing mode can be observed in Fig 2 due to the presence of noise. The diagram shows that extracellular potassium is the bifurcating parameter that leads to the qualitative switches in spiking.

**Experimental manipulation of extracellular potassium.** To experimentally test whether elevated levels of extracellular potassium can induce HOM dynamics of action potential

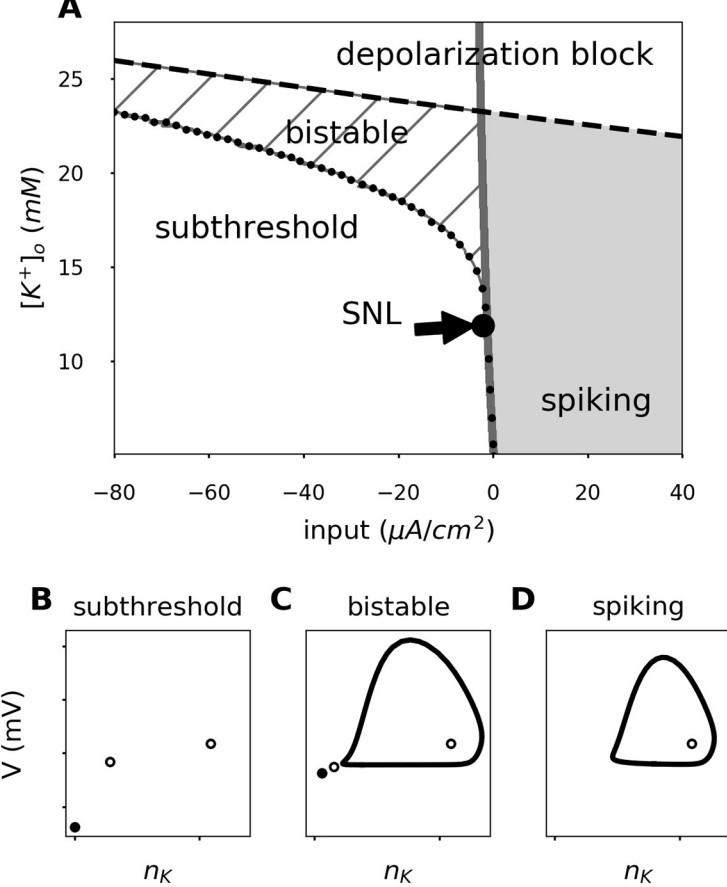

**Fig 3. Characteristic phase portraits in the extracellular potassium / applied current space.** Different combinations of extracellular potassium and input current yield different phase portraits of the fast spike generating sub-system. **A**. The background color represents the characteristic response of that area; Subthreshold or depolarization block: White; Bistable: Dashed; Spiking: Gray. The different regions are separated by the disappearance of the stable node (gray line) and the limit cycle onset (black dots). Examples of phase portraits in each region of the extracellular potassium—input current plane are portrayed; **B**. subthreshold, **C**. bistable, and **D**. spiking state.

generation, a verification of the model-predicted intermittently-interrupted burst-like firing mode suggests itself. In vitro, activity-driven accumulation of extracellular potassium is difficult to reproduce due to the continuously perfused bathing solution that constrains extracellular ion concentrations. We therefore recorded current-induced activity in mouse cortical pyramidal neurons exposed to different fixed concentrations of extracellular potassium. Action potentials were induced by constant-current stimulation in baseline conditions (3 mM extracellular potassium), and after increasing the concentration of extracellular potassium to 10 or 12 mM (see Materials and methods section). Neurons were stimulated with somatic current injection sufficient to maintain the membrane potential close to spiking threshold (see first panel of Fig 4), which in terms of dynamical system analysis, is close to limit cycle onset and for HOM dynamics, also to the bistable region, see Fig 3.

Our experimental results support the model prediction portrayed in Fig 3, in which an increase in extracellular potassium concentration switches the spike generating mechanism. When extracelluar potassium is low (3 mM), the neuron shows very rhythmic (regular) action potential generation over time (see Fig 4 upper panel). In contrast, when extracellular

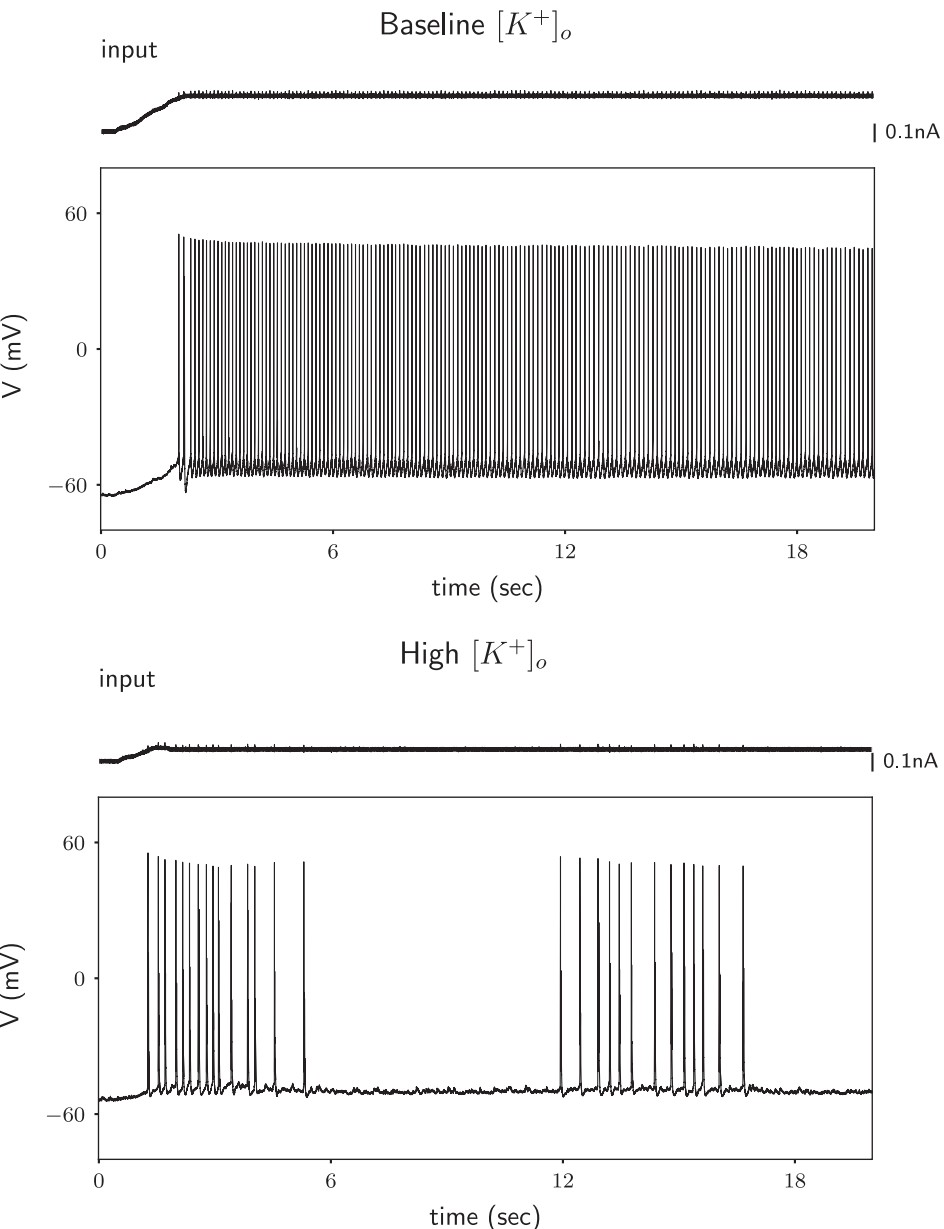

**Fig 4. Rodent cortical neurons exposed to high extracellular potassium show intermittently interrupted firing.**
Example cell; Response of a neuron to a just suprathreshold stimulus in low (3 mM; Top) and high (10 mM; Bottom) extracellular potassium conditions. The suprathreshold current is taken as the current needed to elicit the first spike when injecting a ramp with a shallow slope. The small spikelets visible in the current trace are artifacts resulting from limited capacitive coupling of two channels at the digitizer (i.e., of the action potentials present in the voltage trace), and are not reflective of current injected into the neuron.

potassium is increased to 10 or 12 mM, action potential generation in the same neuron becomes irregular (see Fig 4 lower panel). In 7 out of 8 neurons tested, we observed an increase in spiking irregularity when potassium levels were increased from 3 mM to 12 mM (see supplementary material Figs F and G in S3 Text). In 3 out of 10 neurons tested, we observed an increase in spiking irregularity when potassium levels were increased from 3 mM to 10 mM (see supplementary material Fig I in S3 Text), confirming a crucial prediction of the model. To

rule out that irregularity came from network effects, we repeated the experiment while block-ing synaptic input, and in 8 out of 12 neurons we observed an increase in spiking irregularity (Fig H in S3 Text). We hypothesize that not all neurons exhibited the same qualitative behavior because the distance to the switching point depends on other parameters as well [22] and hence is likely to be variable across cells [23]. It remains to be determined whether the activity of a single neuron generates a sufficient increase in $[K^+]_o$ to induce the SNL bifurcation. Pre-sumably such transitions would be more likely when there are multiple neurons activated simultaneously, which would generate a larger $[K^+]$ flux to the extracellular space.

## Consequences of intracellular sodium accumulation

The dynamical system analysis described above provides a mechanistic explanation for the emergence of intermittently-interrupted firing. However, the only concentration we varied for the analysis was that of potassium ions. Yet long periods of spiking accumulate both extracellu-lar potassium and intracellular sodium ions. In the following section, we therefore describe physiological features that are altered by sodium accumulation. The space of sodium concen-trations is explored to determine whether our results described above regarding extracellular potassium accumulation hold up under conditions of parallel intracellular sodium accumulation.

Sodium accumulation shapes two main properties of spike generation: spike amplitude and spiking threshold, which are determinant features for information transmission and encoding, respectively.

**Intracellular-sodium-dependent spike amplitude reduction.** As outlined above, action potential amplitude is reduced as intracellular sodium accumulates during spiking (Figs 1 and 2), reducing $E_{Na}$ and hence the driving force for sodium current. This effect is also reflected in the phase portraits (Fig 2**C**). The height of the stable limit cycle is squeezed during stimulation, correlating with the $E_{Na}$ reduction (Fig 1**A**). We tested this model prediction in mouse cortical neurons *in vitro* during extended periods of current-induced spiking.

Extended activation of rodent cortical neurons led to a slow spike amplitude reduction (see Supplementary material Fig D in S2 Text). Rodent cortical neurons were activated for 40 sec-onds using short (2 ms) depolarizing current pulses (3 nA) generated at 40 Hz. These neurons exhibited a slow and progressive reduction in spike amplitude that was best fit by a double exponential decay with an average fast time constant ($\tau_{fast}$) of 480 ms and an average slow time constant ($\tau_{slow}$) of 17.7 s (n = 50) (see supplementary material Fig E and Table A in S2 Text for details). The observed slow spike amplitude decay fits the prediction of the model, given that sodium accumulation occurs on the order of seconds, and the faster time scale coincides with the previously reported effects of sodium inactivation [13, 24].

More than one process contributes to spike amplitude decay: a fast process (sodium chan-nel inactivation) and a slow process (sodium accumulation). In order to disentangle the contri-bution of the two, we punctuated prolonged somatic current application with progressively-longer (100 to 1000 ms) breaks sufficient to reset (deinactivate) sodium channels, but not long enough for the Na-K-ATPase to clear activity-dependent increases in intracellular sodium (Fig 5). We observed a progressive reduction in action potential amplitudes that was not rescued by hyperpolarizations as long as one second. Further, the slow time constant of spike ampli-tude decay was found to coincide with that measured in the previous protocol, $\tau_{slow} = 15.7s$ (Fig E, and Tables A and B in S2 Text). These observations suggest that the slow component of amplitude decay cannot result from sodium channel inactivation, and instead is likely driven by intracellular sodium accumulation and the resulting decrease in $E_{Na}$.

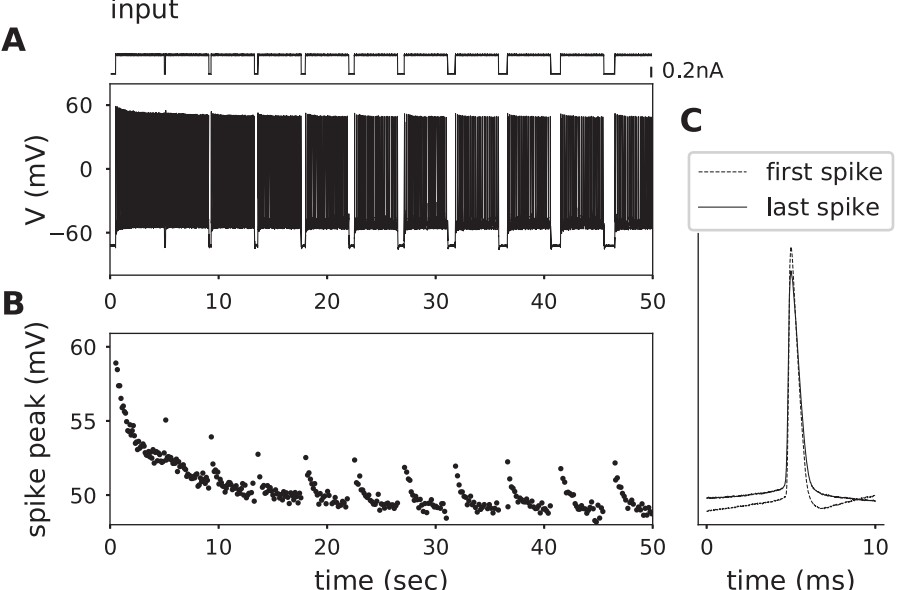

**Fig 5. Activity-dependent decrement in action potential amplitudes. A**. Membrane potential of a rodent cortical neuron (lower trace), subjected to prolonged DC current punctuated by brief hyperpolarizing pulses with different durations (top trace). **B**. Plot of peak voltage for each action potential shown in A. Note that action potential peaks fail to recover after even 1 second long breaks in current stimulation. **C**. Overlay of the first and last action potentials shown in A (aligned at 5 ms).

**Intracellular sodium accumulation shifts the spiking threshold in the model.** Given that an accumulation of intracellular sodium ($[Na^+]_i$) is likely to affect spike generation, we next systematically evaluated the dynamical regimes identified in the bifurcation diagram (Fig 3) at different, fixed $[Na^+]_i$ concentrations.

We found an identical bifurcation structure and splitting into different dynamical regimes for a wide range of $[Na^+]_i$ levels, with the exception of a shift towards higher input currents with larger $[Na^+]_i$ (Fig 6). In other words, as $[Na^+]_i$ accumulates, the spiking threshold is shifted to higher inputs. This shift can be attributed to the dependence of the Na-K pump on $[Na^+]_i$; accumulation of $[Na^+]_i$ thus strengthens the hyperpolarizing pump current, counter-acting the input current and reducing the net excitatory drive. Consequently, the bistable region was shifted along the current axis. Neither a significant change in the area of the bistable region, nor in the location of the transition point towards bistability (i.e., the SNL bifurcation) on the $[K^+]_o$ axis were observed.

## Consequences of simultaneous $[Na^+]_i$ and $[K^+]_o$ changes

The effect of ionic concentrations on neuronal voltage dynamics unfolds via changes in the respective reversal potentials, $E_{Na}$ and $E_K$. Fixing the input current, we can summarize how the spiking regime depends on the concentrations ($[Na^+]_i$ and $[K^+]_o$) in a plot that depicts the spiking regime (reached in the steady-state of the fast subsystem) as a function of the two corresponding reversal potentials (Fig 7). The regime was determined via the phase plane of each system (Fig 3B–3D) and can be classified as bistable, regularly spiking, or stable-resting (i.e., either subthreshold or in depolarization block). To relate the reduced fast subsystem with the complete system including slow concentration dynamics, three example trajectories of the complete system at different initial conditions in the space of reversal potentials are shown on

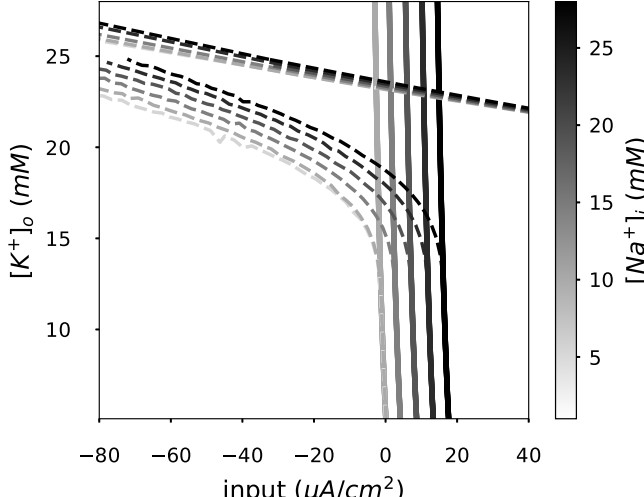

**Fig 6. Extracellular-potassium- and intracellular-sodium-dependent bistable area.** Same bifurcation diagram portrayed in Fig 3 for different intracellular sodium concentrations $[Na^+]_i$. $[Na^+]_i$ controls the input current required to transition from resting to spiking regimes. $[Na^+]_i$ accumulation shifts the bistable region to higher current values.

top of the steady states of the fast subsystem. Each trajectory represents the evolution of ionic concentrations during 10 seconds of stimulation with a fixed current in the presence of noise (as in Fig 2). The corresponding voltage traces are presented for comparison (Fig 7**B**–7**D**). Traces that started at a higher firing rate (and, consequently, were accompanied by larger changes of ionic concentrations) moved farther than the ones that started out at a lower rate.

Which spiking regime a model neuron enters during stimulation can be read off the corresponding trajectory in the complete system. The trajectory depends on the initial ionic concentrations at stimulation onset. A neuron starting with a very low $E_K$ yet high $E_{Na}$ tends to move from regular spiking either to a resting state or to a lower firing rate within the regularly spiking regime (yellow trajectory in Fig 7**D**). Biophysically, spike-frequency adaptation results from the activity of the electrogenic pump, which generates a progressively larger hyperpolarizing current as levels of intracellular sodium increase. Trajectories initialized at low $E_K$ do not reach the bistable region. They tend to a quiescent mode, remaining close to the border to regular firing. If the initial $E_K$ is more elevated, however, a neuron that starts in the regularly spiking regime can reach the bistable region (orange trajectory in Fig 7**C**, similar to the example trace in Fig 2). Very high initial extracellular potassium concentrations promote depolarization block, but, depending on initial conditions, the bistable regime may also be encountered as an intermediate state (magenta trajectory in Fig 7**B**).

The three example trajectories displayed in Fig 7, illustrate that neurons with identical ion channels and stimulation can generate extremely different responses depending on the extracellular environment. Recent spiking activity of neurons alters their response, even when stimulation is unchanged; the rate of change of ionic concentrations strongly depends on neuronal firing rate. Consequently, neurons receiving strong and prolonged stimulation are more likely to experience dynamical regime changes due to ionic accumulation than neurons with weaker stimulation.

## Discussion

In this study, we show that activity-dependent changes in ionic gradients during prolonged neuronal activation can qualitatively change the underlying neuronal dynamics. This is most

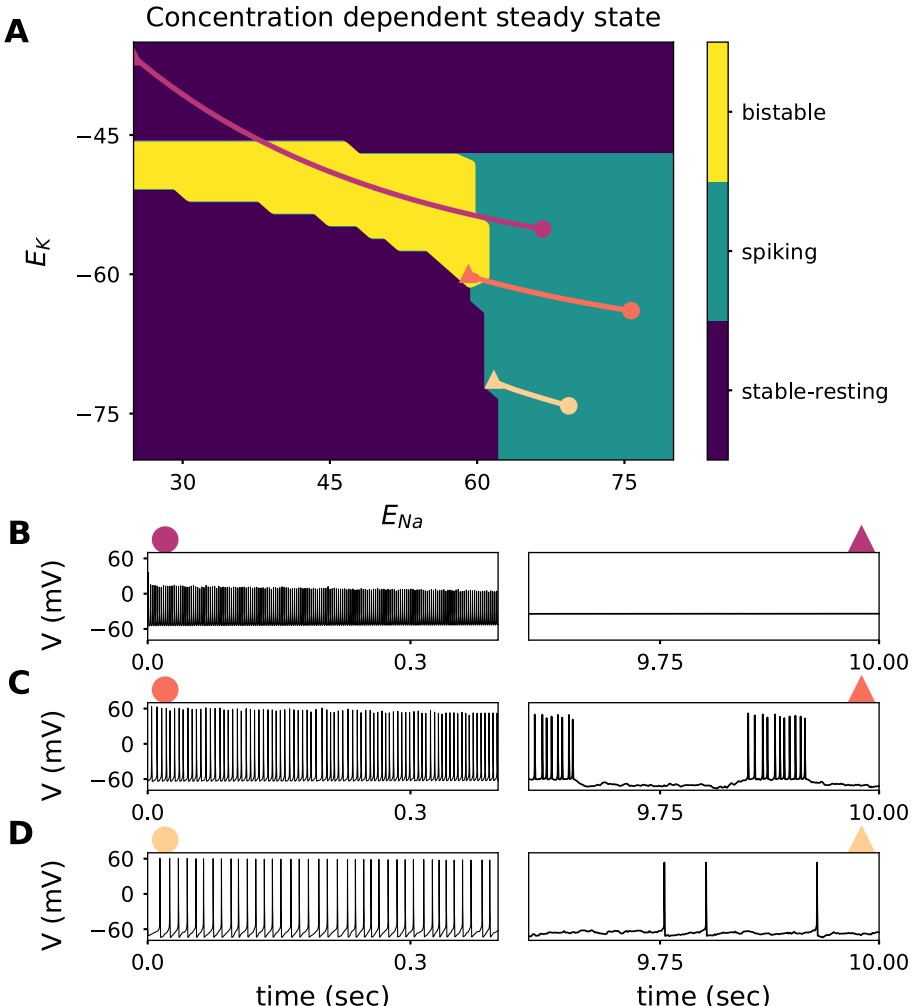

**Fig 7. Consequences of simultaneous $[Na^+]_i$ and $[K^+]_o$ slow dynamics on the fast spike generating dynamics. A**. Characteristic response of the reduced model (receiving a constant input current stimulus of $1\mu A/cm^2$) along the reversal potential plane. The characteristic response can be split in three categories; stable-resting state (purple areas in the lower left corner and top of the graph, representing the subthreshold regime and depolarization block, respectively), the spiking state (green), and the bistable state (yellow). Three example trajectories of the complete system (Including the slow activity-dependent concentration dynamics) simulated during 10 seconds with an irregular input (mean of $1\mu A/cm^2$ and standard deviation of $1\mu A/cm^2$). Initial conditions represented by a circle, and the state of the system 10 seconds later with a triangle. **B**. Membrane potential trace of the trajectory with initial conditions $E_K = -55.1mV$ and $E_{Na} = 66.7mV$ displayed in A. Left panel shows the first 400ms of simulation (marked with a circle), and the right panel shows the last 400ms out of the 10 seconds simulation (marked with a triangle). **C**. Membrane potential trace of the trajectory with initial conditions $E_K = -63.9mV$ and $E_{Na} = 75.7mV$ displayed in A. **B**. Membrane potential trace of the trajectory with initial conditions $E_K = -74.2mV$ and $E_{Na} = 69.3mV$ displayed in A.

apparent when spiking patterns change from regular firing to the intermittently interrupted mode as extracellular potassium accumulates. Intracellular sodium accumulation in contrast mediates a long-lasting spike-frequency adaptation via engagement of the sodium potassium pump and lowers spike amplitude by its effect on the reversal potential $E_{Na}$.

We claim that for highly active neurons, an assumption of stationarity of neuronal dynamics is not precise; neurons are intrinsically affected by their recent electrical activity, beyond any other changes that may arise from network feedback. Prolonged periods of spiking activity

result in modifications of the intracellular and extracellular concentrations of sodium and potassium respectively, and triggers homeostatic mechanisms that regulate ionic gradients at time scales much slower than action potential genesis. The non-stationarity of neurons bears consequences for neural computation.

## Switching to the HOM firing regime

We demonstrate that neuron models that start out with SNIC dynamics (i.e., the classical type I dynamics that has been thought to underlie the firing of most cells with a smooth onset of firing at threshold) can flip to HOM dynamics only seconds after the onset of the spike-inducing stimulation. This transition most obviously manifests in the spiking pattern, which turns from a regular firing mode to an intermittently interrupted one. Our bifurcation analysis shows that an accumulation of extracellular potassium drives this change, instantiating a bistability of the membrane potential. This finding is consistent with previous modelling work that studied the effect of extracellular potassium concentration changes [8, 25–27].

In this particular setting, an increase in the Nernst potential for potassium ($E_K$) and the Nernst potential for the leak ($E_L$) by constraining the magnitude of the potassium ($I_K$) and leak ($I_L$) currents, respectively, shift the spike (limit cycle orbit) towards more depolarized membrane potentials and farther from the resting state (stable node). When both attractors are sufficiently far from each other, both can coexist. The stable node does not lose its stability due to the hyperpolarizing current that is received by the neuron model, either via an external input (Fig 3), or via the hyperpolarizing pump current (Fig 6). Interestingly, other parameters such as temperature and capacitance can also induce the same bistability reported here [22] via a different biophysical mechanism, yet an identical bifurcation structure.

In the presence of a fluctuating input (be it noise or signal), the bistability renders neurons susceptible to switches between the two stable states, giving rise to an irregular, intermittently interrupted firing pattern of short firing phases and pauses of different durations. It has been shown that combining multistable systems, similar to the one reported here, with noise can have unexpected information transmission consequences such as "inverse stochastic resonance" [28], as well as drastic alteration in spiking statistics [29].

Long periods of silence that can be prominent in this mode resemble the ones observed during deterministic bursting reported by [8, 9, 27, 30–33]. In contrast to the deterministic bursting reported by the previous authors, however, the burst-like firing we describe here is driven by the input fluctuations and the bistable state. Thus, neurons in an environment with high extracellular potassium concentration, promoting HOM dynamics, may be more sensitive to input variability.

The SNIC and the HOM regimes yield very different neuronal encoding properties. For instance, the relationship between the input current and the neuronal firing rate (i.e., the gain) depends on the dynamical regime. The gain function of a neuron in the SNIC regime is continuous, the firing rate of such neuron is a continuous function of the input current. The gain function of a neuron in the HOM regime is discontinuous when irregular input is injected, meaning that the firing rate does not smoothly increase as a function of the input current but transitions from no spikes to high frequency spiking abruptly. The phase response curve (PRC), which captures the temporal sensitivity to inputs, also differs between the two regimes. In the SNIC regime, neurons display symmetric PRCs. As PRC symmetry predicts the synchronization of the neurons in the network [22, 34, 35], the switch in firing regime and the underlying bifurcation must also impact the propensity of the neuron to synchronize with other cells in its local network and beyond. Encoding capabilities (such as the profile of frequencies transmitted) are likely to be affected, as they also depend on the PRC characteristics.

Interestingly, intracellular sodium accumulation only quantitatively modulates the qualitative change in spiking regime, underlining the importance of potassium accumulation in this process. The effect of extracellular potassium on neuronal activity has been widely studied [36–38]. Experimental observations have found extracellular potassium dependent bursting [38] and its influence on other dynamical features [25, 26, 39]. While a bistability has been previously observed "in passing" [25, 26], we report a systematic effect and provide mechanistic explanations for the activity-driven changes.

Interpreting these results, we speculate that activity-dependent extracellular potassium accumulation can contribute to, or even induce, epileptiform activity as it has been proposed before [32]. Furthermore, epileptiform activity can be induced by manipulating extracellular potassium concentrations [40]. Both the bursting nature of HOM dynamics [41, 42], as well as their comparatively high susceptability to synchronization in inhibitory networks because of the HOM-characteristic PRC [22], favour synchronized, hyperexcitable states. *In vivo*, Singer and Lux observed that extracellular potassium accumulates in the visual cortex when a rapidly changing visual stimulus is presented to the cat's retina [4]. Remarkably, similar visual stimuli elicit reflex seizures in 4–7% of human epilepsy patients [43]. Reflex seizures could be promoted by extracellular potassium accumulation occurring throughout the visual cortical region that is activated by visual stimulation.

Indeed, the observed drastic consequences of potassium accumulation might occur more frequently *in vivo* than *in vitro*. *In vitro*, extracellular potassium concentrations are clamped. The tissue is perfused with a solution that has a fixed $[K^+]_o$ concentration, analogue to an infinite buffer. *In vivo*, however, extracellular potassium concentration undergoes stimulus-induced changes. In the cat visual cortex, for instance, extracellular potassium accumulates when a graded stimulus is presented to the cat's retina [4]. Regarding the universality of the dynamical changes described here, we expect these to generalize beyond the specific model choice of this study. Specifically, the bistability of conductance based models arises from a slow-down of hyperpolarization, which pushes the limit cycle trajectory to approach the saddle node through a very attractive path (i.e., a strong manifold). This feature can be expected in any neuron model that starts out with SNIC dynamics and ubiquitously favours a switch to HOM dynamics [22, 44]. The exact reversal potential at which the bistability is induced could be shifted by other parameters such as the leak conductance (refer to Fig A in S1 Text). Thus, the exact switching point between the two dynamical regimes can vary between neurons, as their properties are diverse [23, 45]. Therefore, we expect that the exact location of the switching point (i.e., the potassium concentration at which the switch is to be expected) will depend on cellular characteristics, both in neuron models as well as in neurons in vivo. Along these lines, milder extracellular potassium accumulations could suffice to induce the transition in cells with a lower critical value, singling out cells with a higher likelihood to switch their dynamics.

### Attenuation of the spike amplitude

Next to the very prominent change in spiking regime, accumulation of ions are also reflected in the shape of action-potentials, namely the reduction of their peak amplitudes. Such attenuation is a regular feature observed in electrophysiological recordings. It has, however, been previously attributed to inactivation of sodium channels [14, 24]. Our data now suggest that activity-induced changes in the sodium reversal potential contribute substantially to the attenuation of spike peaks, especially during long periods of activity, as they far outlast the effects of inactivation. Our deinactivation experiments with hyperpolarizing current steps support this hypothesis and confirm that the larger and slower component of spike amplitude reduction

persist even when sodium channel inactivation is largely diminished. Moreover, the timescales over which $E_{Na}$ and peak amplitudes are reduced are close to identical for long recordings.

### Concentration-change induced spike-frequency adaptation

Spike-frequency adaptation resulted from an activity-dependent increase in the hyperpolarizing sodium pump current (again mediated by sodium accumulation). This observation was previously reported for leech mechanoreceptor neurons [46] as well as for rodent cortical neurons [5]. The later study [5] demonstrated not only that the pump current produces a slow afterhyperpolarization (AHP) as a consequence of neural activity, but that its time-course mirrors the time-course of intracellular sodium decay. Results from our model are consistent with this finding.

Interestingly, for both spike amplitude attenuation and spike frequency adaptation, ion-channel-mediated equivalent effects on short timescales are well known. The dynamics of concentrations seem to smoothly extend these effects in time.

### Limitations

We note that our model does not consider extracellular uptake of potassium by glial cells. The latter maintain the ionic homeostasis of the extracellular environment and serve as extracellular potassium buffers [47]. Experimental work Dallerac et al. [48] has shown that glial buffering of extracellular potassium saturates when changes are relatively fast. Yet the presence of glial cells *in vivo* is likely to slow the timescale of potassium accumulation. Another aspect that we did not include in the model is spatial ionic diffusion, which implies the assumption of a periodic boundary, meaning that all the neighboring neurons are undergoing the same transition in a synchronized fashion. Thus, including spatial diffusion would slow down and probably also shorten the periods of high accumulation. The effects described here are expected to hold on the order of seconds after activity onset, but given the lack of homeostatic mechanisms, the model does not account for ionic concentration recovery. The predictions presented here can be expected to arise most prominently in pathological conditions, be it glial dysfunction, injury, or energy-deprivation that impairs the pumps and thus facilitates accumulation of both extracellular potassium and intracellular sodium.

### Conclusion

Our analysis shows that a consideration of the slow ionic concentration dynamics inherent to *in vivo* brain activity unravels the nonstationary nature of neurons as computational units. Cortical neurons are typically grouped as either intrinsically bursting, regular spiking, or fast spiking [49]. Here, we reveal a more dynamic situation: by accumulation of ions during prolonged activity, regularly spiking neurons may transition to an intermittently firing mode, or even resemble intrinsically bursting neurons, via activity-induced switches in the underlying bifurcation structure of its dynamics. Neuronal firing patterns are dynamical even in the absence of network changes and strongly depend on the concentrations in the extracellular and intracellular medium. In particular, HOM-type dynamics are likely to be induced in situations of impaired ionic homeostasis, such as glial pathologies or reduced energetic supplies, affecting neural encoding and potentially the network state.

### Materials and methods

With the purpose of understanding the effect of different ionic concentrations on the neuron's response, we used two approaches: simulations of a single-neuron mathematical model with

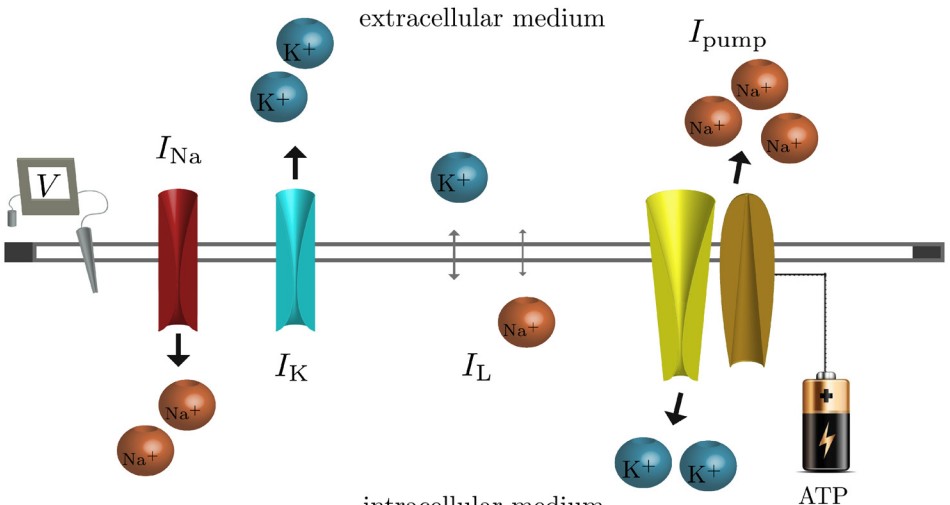

**Fig 8. Summary of the dynamics represented in the model.** The figure illustrates the flux of ions through the fast sodium ($I_{Na}$), the delayed rectifier $I_K$ spike generating currents, the Na-K-ATPase pump, and through the neuronal membrane ($I_L$). The mathematical representation of the currents flowing through proteins can be found in Eqs (2), (3), (8) and (4). The change of the membrane potential due to those currents is represented by Eq (1).

dynamic concentrations, and whole-cell recording of rodent cortical neurons while perfusing the medium to control the extracellular ionic concentrations.

## Ethics statement

Physiological experiments were approved by the Institutional Animal Care and Use Committee of Dartmouth College.

## Computational model

Our goal is to understand the effects of ionic concentration dynamics on the excitability of neurons. Two ingredients are needed: an excitable system (capable of generating spikes) for which we use the Traub-Miles formulation [50], and a description for the slow ionic concentration dynamics. For the excitable system we chose the Traub-Miles model because it is one of the simplest mammalian models with type I dynamics (which display continuous and smooth FI curves). A summary scheme of the spike kinetics (Eqs (1)–(4)) and the concentration dynamics (Eqs (9)–(11)) is depicted in Fig 8. The action potential dynamics at a membrane is governed by a current balance equation involving the following ionic currents,

$$C_{\mathrm{m}}\frac{dV}{dt} = I_{\mathrm{app}} - I_{\mathrm{Na}} - I_{\mathrm{K}} - I_{\mathrm{L}} - I_{\mathrm{pump}}. \tag{1}$$

$$I_{\mathrm{Na}} = g_{\mathrm{Na}} m_{Na}^3 h_{Na}(V - E_{\mathrm{Na}}) \tag{2}$$

$$I_{\mathrm{K}} = g_{\mathrm{K}} n_K^4 (V - E_{\mathrm{K}}) \tag{3}$$

$$I_{\mathrm{L}} = g_L(V - E_{\mathrm{L}}) \tag{4}$$

$$E_K = \frac{RT}{F} \ln\left(\frac{[\text{K}^+]_o}{[\text{K}^+]_i}\right) \tag{5}$$

$$E_{Na} = \frac{RT}{F} \ln\left(\frac{[\text{Na}^+]_o}{[\text{Na}^+]_i}\right) \tag{6}$$

The leak potential ($E_L$) is described as an approximation of the Goldman–Hodgkin–Katz (GHK) equation for the membrane potential;

$$E_L = \frac{RT}{F} \ln\left(\frac{P_K[\text{K}^+]_o + P_{Na}[\text{Na}^+]_o}{P_K[\text{K}^+]_i + P_{Na}[\text{Na}^+]_i}\right). \tag{7}$$

Given that we don't consider chloride dynamics, $I_L = I_{L_K} + I_{L_{Na}}$. Where $I_{L_K} = g_L P_K(V - E_K)$, and $I_{L_{Na}} = g_L P_{Na}(V - E_{Na})$.

$$I_{pump} = \begin{cases} 0 & [\text{Na}^+]_i \leq [\text{Na}]_s \\ \dfrac{I_{maxp}}{1 + \exp(k_{Na}([\text{Na}^+]_i - [\text{Na}]_s))} & [\text{Na}^+]_i > [\text{Na}]_s \end{cases} \tag{8}$$

The pump model in Eq (8) constitutes a homeostatic mechanism that counteracts the movement of ions due to chemical gradients during neuronal spiking activity. Specifically, the sodium potassium pump (Na-K-ATPase) pumps 3 sodium ions out of the cell, while 2 potassium ions enter the cell every pump cycle. The pump is represented by sigmoidal function of the intracellular Na$^+$ concentration in Eq (8) [31, 51]. $I_{maxp}$ is the maximum pump rate which was chosen such that the neuron does not run into depolarization block for a broad range of stimulus intensities. Here, the pump's dependence on potassium and on voltage is ignored. This is an analog of the $\alpha$3 isoform of the Na-K-ATPase, which reacts very strongly to intracellular sodium changes, but is rather insensitive to potassium in the range we study ([K$^+]_o$ 4–20 mM) [17]. $k_{Na}$ is the sodium sensitivity of the pump to sodium, and [Na]$_s$ is the target sodium concentration. Both values were set to 0.1/mM and 20 mM respectively, to match the range of highest pump $\alpha$3 affinity to intracellular sodium [18].

Other modelling studies use pump models that resemble more closely the Na-K-ATPase $\alpha$2 isoform [6, 11, 12]. The main difference between both isoforms is that $\alpha$2 has a much higher sensitivity to extracellular potassium and membrane potential than $\alpha$3 isoform [18]. Developmental studies of isoform expression in the brain have shown that neuronal cells from newborn mice express mainly $\alpha$2, whereas adult animals express mainly $\alpha$3 [52]. And given that our our physiological experiments were performed in neurons from adult mice, we found $\alpha$3 isoform more appropriate.

To understand how our pump choice determines the qualitative results presented here, we performed a bifurcation analysis to the model described above with a pump expression that resembles pump isoform $\alpha$2, refer to Fig C in S1 Text. The main difference between the pump expression in Eq 8 and the one used to resemble $\alpha$2 (Fig C in S1 Text), is the pump sensitivity to extracellular potassium. Fig C in S1 Text illustrates how the pump sensitivity to $[K^+]_o$ shifts the spiking threshold towards higher input currents as $[K^+]_o$ accumulates.

The concentration dynamics is influenced by the transmembrane currents due to ion channels and the pump as follows.

$$\frac{d[\mathrm{Na^+}]_i}{dt} = \frac{\rho}{F}\left(-3I_{\mathrm{pump}} - I_{\mathrm{Na}} - I_{L_{Na}}\right) \tag{9}$$

$$\frac{d[\mathrm{K^+}]_i}{dt} = \frac{\rho}{F}\left(2I_{\mathrm{pump}} - I_{\mathrm{K}} - I_{L_K}\right) \tag{10}$$

$$\frac{d[\mathrm{K^+}]_o}{dt} = -\frac{d[\mathrm{K^+}]_i}{dt}\frac{\mathrm{Vol}_i}{\mathrm{Vol}_e} \tag{11}$$

Where $\rho$ represents the surface to volume area, and here is set to 4000/cm [31], $\frac{Vol_i}{Vol_e}$ is set to 0.2 (note that this is a very small value mainly because we are not including extracellular regulation mechanisms, thus we assume glial cells, blood vessels, myelin, and other structures are part of the extracellular volume. Previous experimental measurements yield a $\frac{Vol_i}{Vol_e}$ of approximately 0.4 [53], however we decreased it to compensate for the lack of other homeostatic mechanisms and allow reproducibility of experimental observations. All expressions and parameters used in the simulations can be found in Tables C-F in S4 Text.

**Time scale separation.** Simulating the model dynamics with an ODE solver is very time consuming (this was done for voltage traces in Figs 1, 2 and 7). Therefore, in order to characterize the system's response to a broad set of initial conditions using shorter simulation times, we used time scale separation for the analysis. This technique is particularly useful for our set of equations because the system contains variables changing in very slow and very fast time scales.

Ionic concentration dynamics change with a time scale in the order of seconds, while spike generating currents are changing in the order of milliseconds. Thus, we can split the system into two subsystems. The fast subsystem (Eqs 1, 2, 3, 4, 7 and 8) receives ionic concentrations ($[\mathrm{Na^+}]_i$, $[\mathrm{K^+}]_i$, $[\mathrm{K^+}]_o$,) as fixed parameters. For each parameter combination the steady states are portrayed in phase portraits (See Fig 2 bottom panel). A parameter combination that yields a phase portrait containing only one stable node is characterized as resting state, one containing only a stable limit cycle is characterized as regular firing, and one with a stable node and a stable limit cycle is characterized as bistable (See Fig 7A).

**Bifurcation analysis.** The numerical bifurcation software AUTO [54] was used to find the limit cycle onset (spike onset), the disappearance of the steady state, and the hopf bifurcation (depolarization block). The analysis was repeated for different ionic concentrations (See in Figs 3 and 6). First a particular combination of ionic concentrations is chosen and the steady states of the dynamical system with the combination of parameters are calculated with a regular ODE, secondly the applied current is used as the continuation parameter. AUTO, detects the stable/unstable nodes and limit cycles, it also points where the stability of such objects changes (bifurcations). For this work the most relevant ones are; the points where the stable node transitions from being stable to unstable (saddle-node bifurcation), the point where a stable limit cycle orbit emerges (HOM), and the point where the stable limit cycle disappears (Hopf bifurcation). Such points can be continued along all the model parameters efficiently in AUTO. Fig 6 is constructed by continuing the 3 points mentioned above, in the extracellular potassium—applied current plane.

## Experimental protocol

Physiological experiments were approved by the Institutional Animal Care and Use Committee of Dartmouth College. Female and male adult (3- to 4-month-old) C57BL/6J mice were bred in facilities accredited by the Association for Assessment and Accreditation of Laboratory Animal Care and maintained on a 12h-12h light-dark cycle with continuous free access to food and water.

On the day of experiments, mice were anesthetized with vaporized isoflurane and decapitated, with brains rapidly removed into an artificial cerebral spinal fluid (aCSF) composed of (in mM): 125 NaCl, 25 $NaHCO_3$, 3 KCl, 1.25 $NaH_2PO_4$, 0.5 $CaCl_2$, 6 $MgCl_2$ and 25 glucose (saturated with 95% O2–5% $CO_2$). Coronal brain slices (250 $\mu m$ thick) of the frontal cortex were cut using a Leica VT 1200 slicer and stored in a holding chamber filled with aCSF containing 2 mM $CaCl_2$ and 1 mM $MgCl_2$. Slices were maintained in the holding chamber for 45 minutes at 35˚C, and then at room temperature ($\sim 25$˚C) until use in experiments.

Slices were transferred to a recording chamber on a fixed-stage microscope (Olympus), and continuously perfused ($\sim$ 7 ml/min) with oxygenated aCSF heated to 35–36˚C. Layer 5 pyramidal neurons in the prelimbic cortex were visually targeted using a 60X water-immersion objective, and whole-cell recordings made with patch pipettes (5–7 M$\Omega$) filled with a solution containing the following (in mM): 135 potassium gluconate, 2 NaCl, 2 $MgCl_2$, 10 HEPES, 3 $Na_2ATP$ and 0.3 $Na_2GTP$, pH 7.2 with KOH. Data were acquired using a BVC-700 amplifier (Dagan Corporation) connected to a HEKA 8+8 digitizer or an ITC-18-USB digitizer driven by AxoGraph software (AxoGraph Scientific; RRID: SCR—014284). Membrane potentials were sampled at 50 to 100 kHz and filtered at 5 or 10 kHz. Voltage measurements were corrected for a +12 mV liquid junction potential. Concentrations of KCl (3, 10, or 12 mM) and NaCl (125, 118, or 116 mM, respectively) were adjusted as indicated to test the impact of extracellular potassium concentration on action potential dynamics. In some experiments fast synaptic transmission was blocked with continuous bath-application of the AMPA receptor blocker DNQX (25 $\mu$M; Tocris), the NMDA receptor blocker D-AP5 (25 $\mu$M; Tocris), and the $GABA_A$ receptor blocker picrotoxin (50 $\mu$M; Tocris).

## Data analysis

Data analysis was done in python.

**Spiking irregularity.** Spiking irregularity is measured as

$$CV = \frac{\sigma}{\mu}, \tag{12}$$

where $\sigma$ is the standard deviation of the interspike interval (ISI), and $\mu$ is the mean. In the experiment in which synaptic input was blocked we excluded cells that showed patterns different from regular type I neurons in the baseline condition to rule out pathological firing ($CV > 0.5$).

**Time scale of spike amplitude decay.** The fast and the slow components of the spike amplitude decay were calculated by fitting the time dependent spike-voltage-peak to a double exponential function,

$$D_{\text{fast}} \exp \left[ -\frac{t}{\tau_{\text{fast}}} \right] + D_{\text{slow}} \exp \left[ -\frac{t}{\tau_{\text{slow}}} \right] + D_{\text{ss}}. \tag{13}$$

The distribution of the parameters that yield the best fit across all traces measured are shown in Fig E in S2 Text, and Tables A and B in S2 Text.

## Supporting information

**S1 Fig. Transition from rest to spiking (limit cycle onset bifurcations) for different extracellular potassium concentrations.** From bottom to top; SNIC (saddle-node on invariant circle): Purple, SNL (Saddle-node-loop): Blue; HOM (saddle homoclinic orbit): Green. In the SNIC regime the stable node collides with an unstable node, giving rise to a saddle node. The limit cycle orbit passes through the saddle node, the trajectory leaves the saddle node along the semi-stable manifold. After one period trajectory approaches the saddle node along the same semi-stable manifold. At the SNL point, trajectories leave the saddle node along the semi-stable manifold as in the SNIC case, but after one period those trajectories approach the saddle node along the strongly stable manifold. Notice that the SNL orbit is smaller than the SNIC orbit, and has a shorter period. In the HOM regime a stable node and a limit cycle coexist. External perturbations shift the state of the system from the stable node to the attraction domain of the limit cycle attractor.
(TIFF)

**S1 Text. Fig A**. **Changes in the conductance of the delayed rectifier potassium current ($g_K$) distorts the bistable region portrayed in** [Fig 3A]. Same bifurcation diagram portrayed in [Fig 3A] for different $g_K$. Here the curves correspond to the delayed rectifier conductance of $g_K$; 100, 200, 300, and 340 $msiemens/cm^2$. As $g_K$ increases, the limit cycle onset, and the depolarization block lines are shifted towards higher extracellular potassium concentrations. **Fig B. Changes in the leak conductance ($g_L$) distorts the bistable region portrayed in** [Fig 3A]. Same bifurcation diagram portrayed in [Fig 3A] for different $g_L$. Here the curves correspond to leak conductances of $g_L$; 0.01, 0.1, 0.5, and 1.0 $msiemens/cm^2$. As $g_L$ increases, the bistable region is shifted towards higher extracellular potassium concentrations. Another effect of more leaky neurons, is that the dependence of the spiking threshold on extracellular potassium is more prominent. **Fig C. Extracellular potassium and $[K^+]_o$ pump's sensitivity ($K_{sens}$) dependent bistable area**. Same bifurcation diagram portrayed in [Fig 6] for different $[K^+]_o$ pump's sensitivity. Here 0,0.1,0.2 and 0.5 $1/mM$ sensitivities to $[K^+]_o$ ($K_s$) are portrayed and $[K^+]_s$ is fixed to 4mM for all curves, the expression of the pump that was used here resembles isoform $\alpha_2$ (eq A in S1 Text). $K_s$ distorts the saddle node bifurcation line, curving it towards more depolarized currents, i.e., shifting the spiking threshold towards higher input currents.
(PDF)

**S2 Text. Fig D**. **Slow decay of spike amplitudes**. Voltage recording of a neuron experiencing depolarizing pulses applied at 40 Hz. The fast and slow time constants of amplitude decay were $\tau_{fast} = 410(ms)$ and $\tau_{slow} = 13.6(sec)$, respectively. Notice that the peak of the last spike fails to recover to the initial amplitude after the one-second-long hyper-polarizing pulse. **Fig E. Distribution of time scales of the double exponential decay ([Eq 13]) of the spike amplitude**. Two protocols were used to measure the time scales of spike amplitude decay, an example of the "40 Hz Depolarizations" is shown in Fig D in S2 Text, and an example of the "Hyperpolarization" is shown in [Fig 5]. Notice that the distribution of $\tau_{slow}$ is independent of the protocol used. **Table A. Summary of the distribution of the best fit of the parameters for each of the 50 traces**. Depolarizing pulses applied at a 40Hz rate. **Table B. Summary of the distribution of the best fit of the parameters for each of the 73 traces**. Hyperpolarizing pulses.
(PDF)

**S3 Text. Fig F. Spiking variability calculated as the coefficient of variation ($CV = \frac{\sigma}{\mu}$) for all cells sampled, when stimulating with white noise added to the baseline input.** An increase from 3mM to 12mM in extracellular potassium increased the spiking variability of 5 out of 6

cells measured. **Fig G. Spiking variability calculated as the coefficient of variation** ($CV = \frac{\sigma}{\mu}$) **for all cells sampled, when stimulating with baseline input**. An increase from 3mM to 12mM in extracellular potassium increased the spiking variability of 2 out of 2 cells measured. The main source of stimuli irregularity was the network activity. **Fig H. Spiking variability calculated as the coefficient of variation** ($CV = \frac{\sigma}{\mu}$) **for all cells sampled after blocking synaptic input, under baseline input stimulation**. An increase from 3mM to 10mM in extracellular potassium increased the spiking variability of 8 out of 12 cells measured. **Fig I Spiking variability calculated as the coefficient of variation** ($CV = \frac{\sigma}{\mu}$) **for all cells sampled**. An increase from 3mM to 10mM in extracellular potassium increased the spiking variability of 3 out of 10 cells measured. The main source of stimuli irregularity was the network activity. (PDF)

**S4 Text. Table C. Gating dynamics used for the excitable portion of the model**. **Table D. Expressions used for the excitable portion of the model**. **Table E. Parameters used for the excitable portion of the model**. **Table F. Parameters used for the ionic concentration dynamics portion of the model**. (PDF)

## Acknowledgments

We thank Jens-Steffen Scherer, Jiameng Wu, and Pia Rose for valuable feedback on the manuscript.

## Author Contributions

**Conceptualization:** Susana Andrea Contreras, Jan-Hendrik Schleimer, Susanne Schreiber.

**Data curation:** Susana Andrea Contreras, Allan T. Gulledge.

**Formal analysis:** Susana Andrea Contreras, Jan-Hendrik Schleimer.

**Funding acquisition:** Allan T. Gulledge, Susanne Schreiber.

**Investigation:** Susanne Schreiber.

**Methodology:** Susana Andrea Contreras, Jan-Hendrik Schleimer, Allan T. Gulledge, Susanne Schreiber.

**Project administration:** Susanne Schreiber.

**Supervision:** Jan-Hendrik Schleimer, Susanne Schreiber.

**Writing – original draft:** Susana Andrea Contreras.

**Writing – review & editing:** Jan-Hendrik Schleimer, Allan T. Gulledge, Susanne Schreiber.

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
