## [Decision Letter · Decision Letter 0]

5 Jan 2021

Dear Mrs. Contreras Ceballos,

Thank you very much for submitting your manuscript "Activity-mediated accumulation of potassium induces a switch in firing pattern and neuronal excitability type" for consideration at PLOS Computational Biology.

As with all papers reviewed by the journal, your manuscript was reviewed by members of the editorial board and by several independent reviewers. In light of the reviews (below this email), we would like to invite the resubmission of a significantly-revised version that takes into account the reviewers' comments.

Note that a large number of major concerns have been raised.  These should be thoroughly addressed.  As part of this revision effort, please be sure to cite previous literature appropriately and to provide a clear explanation of where the novelty in this work lies relative to past works; a convincing case on this point will be essential for the review process of the revised manuscript.  Better justification of model and parameter value choices and of robustness of results to these choices will also be essential.

We cannot make any decision about publication until we have seen the revised manuscript and your response to the reviewers' comments. Your revised manuscript is also likely to be sent to reviewers for further evaluation.

Sincerely,

Jonathan Rubin

Associate Editor

PLOS Computational Biology

Kim Blackwell

Deputy Editor

PLOS Computational Biology

Reviewer's Responses to Questions

**Comments to the Authors:**

Reviewer #1: In this manuscript, Contreras et al. investigate how activity-dependent dynamics in the extracellular potassium and intracellular sodium concentrations affect the dynamics of action potential generation in a computational model. This study highlights the importance of considering the often-overlooked role of ion plasticity in shaping neuronal spiking dynamics. The authors demonstrate that activity-induced short-term ionic dynamics can induce changes in firing mode, spike height, spike frequency adaptation, and excitability type. Additionally, the authors provide experimental evidence supporting their findings. Overall, I am enthusiastic about this work but have some concerns.

Major Concerns:

1. The leak current conducts both Na+ and K+ ions. Therefore, in Eq. 7 only a portion of IL should contribute to potassium concentration dynamics. Since IL conducts IK the leak reversal potential should also be dynamic, however, it is not clear from the main text if this is the case.

2. The model of the Na-K-ATPase pump used here does not include any activation dependence on potassium or voltage. It should be discussed how more complicated pump activation dynamics might affect these findings.

3. The experimental data showing a switch from a regular to an intermittent firing pattern would be more convincing with additional neurons and if synaptic transmission were blocked.

4. The bifurcation analysis presented here clearly shows how increasing extracellular potassium leads to a bistable state by the creation and coexistence of a stable fixed point and a stable limit cycle. However, a biophysical explanation could make the findings accessible to a broader audience and provide additional intuition into the underlying dynamics. More specifically, how does depolarizing EK and EL (by increasing extracellular K+) affect the balance of INa, IK and IL resulting in the creation of a stable limit cycle without destroying the stable node?

5. Similarly, how robust is the bistable region presented in Fig. 3A and Fig. 6 to changes in model parameters? For example, if gK and/or gL is increased/decreased does this affect the size and location of the bistable region? Manipulation of extracellular potassium ultimately affects the magnitude of IK and IL. Therefore, gK and gL should affect the bistable region.

Minor Concerns:

1) In the model spike frequency adaptation occurs due to Na+ accumulation and the subsequent outward current generated by the Na/K pump. Typically, increasing extracellular K+ increases excitability and firing rate by depolarizing EL and EK. Would spike-frequency adaptation still occur if gL and/or gK were larger?

2) The orange trajectory in Fig. 7A is said to reach the bistable region, however it looks like the full trajectory remains in the green spiking region.

3) References to supplemental figures do not have figure numbers. See line 197, 224 and 238 for example.

4) Line 368 is missing “of” at the end of the sentence between “effects” and “inactivation”

5) Eq. 5 label should be right justified like the rest of the equations.

Reviewer #2: Review of manuscript PCOMPBIOL-D-20-02004: “Activity-mediated accumulation of potassium induces a switch in firing pattern and neuronal excitability type”.

The authors present a single compartment neural model, including standard ion channels (Na+ & K+) for AP generation, leak currents and a Na/K-pump. They model the ion concentrations for Na+ (intracellularly) and K+ (intra- and extracellularly). The modeling work seems appropriately conducted. As the main frontier of the computational neuroscience field (HBP and so) typically neglects effects of ion concentration dynamics altogether, I am happy to see models that do not.

The model is used to study the consequences that varying ion concentrations has on the firing property of neurons. Some of the key findings are validated through experiments. The perhaps most interesting finding is that extracellular potassium accumulation switches the neuron into a bistable regime described through a to a homoclinic orbit bifurcation (HOM), where it produces intermittently interrupted (burst-like) output.

C1: Although, as mentioned, the standard within the field is to neglect effects of varying ion concentration altogether, there is quite a number of computational works addressing the topic, often in simple single compartment neuron models such as the one presented her. Some papers that seems to be close in topic to the here submitted work are:

Kager, H., Wadman, W. J., & Somjen, G. G. (2000). Simulated seizures and spreading depression in a neuron model incorporating interstitial space and ion concentrations. Journal of neurophysiology, 84(1), 495-512.

Cressman, John R., et al. "The influence of sodium and potassium dynamics on excitability, seizures, and the stability of persistent states: I. Single neuron dynamics." Journal of computational neuroscience 26.2 (2009): 159-170.

Øyehaug, L., Østby, I., Lloyd, C. M., Omholt, S. W., & Einevoll, G. T. (2012). Dependence of spontaneous neuronal firing and depolarisation block on astroglial membrane transport mechanisms. Journal of computational neuroscience, 32(1), 147-165.

Owen, J. A., Barreto, E., & Cressman, J. R. (2013). Controlling seizure-like events by perturbing ion concentration dynamics with periodic stimulation. PLoS One, 8(9), e73820

Hübel, N., Schöll, E., & Dahlem, M. A. (2014). Bistable dynamics underlying excitability of ion homeostasis in neuron models. PLoS Comput Biol, 10(5), e1003551.

Wei, Y., Ullah, G., & Schiff, S. J. (2014). Unification of neuronal spikes, seizures, and spreading depression. Journal of Neuroscience, 34(35), 11733-11743

Sætra, M. J., Einevoll, G. T., & Halnes, G. (2020). An electrodiffusive, ion conserving Pinsky-Rinzel model with homeostatic mechanisms. PLoS computational biology, 16(4), e1007661.

I was surprised to see that neither of these were cited in the manuscript, and I am a bit uncertain as to how the current paper and its conclusions stand out as compared to these previous works. For example, the works by Cresman et al (2009), Øyehaug et al. (2012), Hübel et al (2014) and Wei et al. (2014) all present some bifurcation analysis relating neuronal firing properties to some model variables or parameters, and extracellular K+ play an important role in most of these. Probably, there is something new to the kind of bistability observed in the submitted manuscript relative to the different firing regimes reported previously, but I think a comparison should be made in this regard.

C2: Regarding the model:

- I do not have the book where the Traub-Miles model (used in the current work) is described. I would like to ask why this model was selected for the study. As far as I understand, it is supposed to represent a hippocampal neuron, while the experimental validation was based on a cortical neuron.

- Do you think that your conclusions regarding the bistability regime is sensitive to the choice of model, or do you picture it to be of general validity, i.e., be true for, say, the Hodgkin-Huxley model or any other existing model including the standard Na+ and K+ channels.

- As indicated by the formulation (line 422-424) “Two ingredients are needed: an excitable system (capable of generating spikes) for which we use the Traub-Miles formulation [40] (See equations (1),(2),(3),(4)), and a description for the ionic concentration changes that occur due to ionic currents”, the original Traub-Miller model did not include ion concentration dynamics. Hence, some of the parameters in Table S5 were not from the Traub-Miller model, but new to this paper. I think, then, that it would be appropriate to comment on how these were chosen.

o The values for the parameters Vol_i and Vol_e (or the ratio between them) were not given in the paper.

o As in most models of this kind, the extracellular space was confined, and K+ would accumulate locally without being allowed to “diffuse” away. I suppose it could be added to the discussion that this is equivalent to having periodic boundary conditions (if all neurons in the neighborhood do the same, all extracellular space receive the same amount of K+, so there will be no concentration gradients, and thus no extracellular diffusion).

o The parameter rho had the value 4/cm. Since “rho/F” converts a membrane current density to an intracellular concentration change, rho should interpret as the surface to volume ratio of the neuron. Picturing a spherical neuron, that would give us: rho = 4 pi r^2/(4/3 pi r^3) = 3/r. For rho to have the value 4/cm, r must be ¾ cm, which is a very huge neuron. Any comments on that?

C3: Regarding the presentation:

I would like to find a summarizing figure in the beginning of the results section that illustrates everything that goes into the model (the included membrane mechanisms and key variables, Vm, Na+ and K+).

C4: The model should be made available online.

Reviewer #3: The authors present a neuronal model with standard HH conductances with ionic dynamics including an electrogenic pump. The authors perform a fast slow analysis to investigate the role of ionic dynamics in the behavior of a single neuron and identify the dynamical structure underlying a number of possible neuronal states. The work is interesting and the results are promising, but there are a number of issues that need to be addressed before consideration for publication.

The fast slow analysis described here is similar to previous work by Owen et al (2013) Controlling Seizure-Like Events by Perturbing Ion Concentration Dynamics with Periodic Stimulation. PLOS ONE 8(9): e73820. The role of sodium and potassium dynamics in setting the state of the cell is described in that work. The relation to this work should be addressed.

One interesting effect described by the author is the multi-stable dynamics supported by their model. This multi-stability and switching is a manifestation of the well known inverse stochastic resonance. Please see this reference for details. Uzuntarla et al. Dynamical structure underlying inverse stochastic resonance and its implications Phys. Rev. E 88, 042712 – Published 31 October 2013. This point should be addressed.

The bi-stable state however is not convincingly motivated by the experimental results. The potassium concentrations used are very high and one might expect seizures. Under these conditions experimentally, (Jensen and Yaari. 1997, Role of intrinsic burst firing, potassium accumulation, and electrical coupling in the elevated potassium model of hippocampal epilepsy. J. Neurophys.), and computationally ( Cressman and Ullah et al. The influence of sodium and potassium dynamics on excitability, seizures, and the stability of persistent states: I & II. Network and glial dynamics. J. Computational Neuroscience. 2009.) , using a similar method to what is performed here. These computational models also demonstrated multi stability due to ionic dynamics. How do your results relate to these previous findings?

I am not convinced that 2 out of 2 cells having a higher variability without blocking synapses with high potassium demonstrates an intrinsic cellular response.

Line 72 Why α3? This seems like a strange pump to choose, and I would expect it to play a key role in the sodium potassium dynamics of the cell. This choice needs to be better motivated.

Line 78. This effect has been seen before with other models with ionic dynamics. Somjen et al

Kager, et al, 2007,Seizure-like after discharges simulated in a model neuron. J. of Comp. Neuro)

minor items:

Line 61 slow AHP resulted from the hyperpolarizing Na-K-ATPase pump current:

Did you check to see if the current was responsible for the hyperpolarization, or just the sodium driven decrease in potassium concentrations? Your pump is not sensitive to potassium so it could get very low outside driving your Ek and Vm very negative.

Line 134. Typo

Line 135, permanently probably should be continuously.

Section starting on line 149. How were the different bifurcations identified?

Figure 2. The resting state is below the potassium reversal for the quiescent state. This indicates some tonic negative input. What is the source of this input?

Line 504, seg->ms

Line 508, figure number missing

**Have all data underlying the figures and results presented in the manuscript been provided?**

Reviewer #1: None

Reviewer #2: **No: **The model code should be made available for online download.

Reviewer #3: Yes

PLOS authors have the option to publish the peer review history of their article (what does this mean?). If published, this will include your full peer review and any attached files.

Reviewer #1: No

Reviewer #2: **Yes: **Geir Halnes

Reviewer #3: No
---

## [Decision Letter · Decision Letter 1]

16 Apr 2021

Dear Dr. Schreiber,

We are pleased to inform you that your manuscript 'Activity-mediated accumulation of potassium induces a switch in firing pattern and neuronal excitability type' has been provisionally accepted for publication in PLOS Computational Biology.

Best regards,

Jonathan Rubin

Associate Editor

PLOS Computational Biology

Kim Blackwell

Deputy Editor

PLOS Computational Biology

Reviewer's Responses to Questions

**Comments to the Authors:**

Reviewer #1: I only have one minor comment. The authors have updated equations 9 and 10 for [Na]i and [K]i dynamics to include a portion of the leak conductance carried by Na+ and K+ (I_LNa and I_LK where IL=I_LK+I_LNa). It's not clear how I_LK and I_LNa are defined.

Reviewer #2: The authors have addressed my concerns and responded appropriately. I recommend that this article is accepted.

Reviewer #3: The authors have addressed my main concerns and I recommend the manuscript for publication.

It is a bit of a different set up, but the effects of ionic dynamics on network stability and multistable behavior has been observed previously. The authors might want to have a look.

J Comput Neurosci (2009) 26:171–183

DOI 10.1007/s10827-008-0130-6

**Have the authors made all data and (if applicable) computational code underlying the findings in their manuscript fully available?**

Reviewer #1: None

Reviewer #3: Yes

PLOS authors have the option to publish the peer review history of their article (what does this mean?). If published, this will include your full peer review and any attached files.

Reviewer #1: No

Reviewer #2: **Yes: **Geir Halnes

Reviewer #3: No

**Have all data underlying the figures and results presented in the manuscript been provided?**

Reviewer #2: Yes

---

## [Editor Report · Acceptance letter]

22 May 2021

PCOMPBIOL-D-20-02004R1 

Activity-mediated accumulation of potassium induces a switch in firing pattern and neuronal excitability type

Dear Dr Schreiber,

I am pleased to inform you that your manuscript has been formally accepted for publication in PLOS Computational Biology. Your manuscript is now with our production department and you will be notified of the publication date in due course.

With kind regards,

Andrea Szabo
